# Employing 11-Ketotestosterone as a Target Analyte for Adrenosterone (11OXO) Administration in Doping Controls

**DOI:** 10.3390/metabo14030141

**Published:** 2024-02-26

**Authors:** Thomas Piper, Gregor Fußhöller, Mario Thevis

**Affiliations:** 1Center for Preventive Doping Research, Institute of Biochemistry, German Sport University Cologne, Am Sportpark Müngersdorf 6, 50933 Köln, Germany; 2European Monitoring Center for Emerging Doping Agents (EuMoCEDA), 50933 Köln, Germany

**Keywords:** adrenosterone, 11-ketotestosterone, carbon isotope ratios, urinary concentration threshold, doping controls

## Abstract

Adrenosterone (Androst-4-ene-3,11,17-trione, 11OXO) is forbidden in sports according to the Prohibited List of the World Anti-Doping Agency. The administration of 11OXO may be detected by monitoring the urinary concentrations of its main human metabolites 11β-hydroxy-androsterone and 11β-hydroxy-etiocholanolone. Preliminary urinary concentration and concentration ratio thresholds have been established for sports drug testing purposes, but adaptations are desirable as the suggested limits would result in numerous suspicious findings due to naturally elevated concentrations and ratios. Recently, the metabolism of 11-oxo-testosterone (KT) was investigated in the context of anti-doping research, resulting in a preliminary urinary concentration threshold and a confirmation procedure based on the determination of carbon isotope ratios (CIRs). Gas chromatography coupled to isotope ratio mass spectrometry was employed to investigate the CIRs of selected steroids. As KT is also a metabolite of 11OXO, the developed protocols for KT have been tested to elucidate their potential to detect the administration of 11OXO after a single oral dose of 100 mg. In order to further improve the analytical approach, the threshold for urinary concentrations of KT was re-investigated by employing a reference population of *n* = 5232 routine doping control samples. Quantification of urinary steroids was conducted by employing gas chromatography coupled to triple quadrupole mass spectrometry. Derived from these, a subset of *n* = 106 samples showing elevated concentrations of KT was investigated regarding their CIRs. By means of this, potentially positive samples due to the illicit administration of 11OXO or KT could be excluded, and the calculation of reference population-derived thresholds for the concentrations and CIR of KT was possible. Based on the results, the urinary concentration threshold for KT is suggested to be established at 130 ng/mL.

## 1. Introduction

Adrenosterone (Androst-4-ene-3,11,17-trione, 11OXO) represents an endogenous steroid in the human metabolism of the adrenal glands and emerges in the metabolic pathways of corticosteroids and androst-4-ene-3,17-dione [1,2,3]. The human metabolism of 11OXO has already been addressed, in the 1950s and 1960s, and the main urinary metabolites were identified as 11β-hydroxy-androsterone (OHA), 11β-hydroxy-etiocholanolone (OHE), 11-oxo-androsterone (KA), and 11-oxo-etiocholanolone (KE); all chemical structures are depicted in the Appendix A [4,5]. All of these metabolites can be found in urine at varying concentration levels due to diurnal variations, confounding factors like stress, and large inter-individual variability in general [6,7].

In the context of doping controls, 11OXO belongs to the compounds forbidden for athletes at all times by the World Anti-doping Agency (WADA), and it is explicitly mentioned on WADA’s Prohibited List [8]. A potential approach to detect the misuse of 11OXO was already published in 2009, relying on the urinary concentrations of OHA and the ratio of OHA/OHE [9]. Preliminary thresholds for the initial testing procedure (ITP) were based on a relatively small reference population of *n* = 85 athletes, and it was suggested to forward suspicious samples to isotope ratio mass spectrometry (IRMS)-based investigations to elucidate the carbon isotope ratios (CIRs) of OHA for confirmation purposes. In parallel to the approach for the detection of testosterone misuse, CIRs can be employed to identify the source of urinary steroids, i.e., endogenous production or illicit administration. This method assumes that exogenous steroids usually show depleted CIRs compared to endogenous steroids. CIRs are usually expressed as δ-values against the international standard Vienna Pee Dee Belemnite (*VPDB*) based on Equation (1) [10]:(1)δ13CVPDB=C13C12SAMPLEC13C12VPDB−1

In order to evaluate the measured CIRs, differences between endogenous reference compounds (ERCs), i.e., steroids not affected by the administration, and target analytes (TCs) are considered. The differences found between ERCs and TCs are given as Δ-values as shown in Equation (2):(2)$$∆\left[‰\right]=|\delta^{13}C_{ERC}-\delta^{13}C_{TC}|$$

If the difference between an ERC and a TC exceeds applicable thresholds, the source of the TC is considered to be exogenous.

Recently, the potential detection of the administration of a structurally closely related steroid to 11OXO, 11-oxo-testosterone (KT), was studied [11]. The most promising marker for the detection of KT administration employing routine initial testing procedures (ITPs) is the urinary concentration of KT itself followed by IRMS determinations of either KT or its metabolites (OHA, OHA, KA, KE, and 11-oxo-dihydrotestosterone (KDHT)). Similar to the shortcomings described for 11OXO, the employed reference population to estimate a urinary threshold was relatively small (*n* = 220) and no population-derived thresholds for potential Δ-values were established.

Based on these findings, the following research objectives were addressed in the presented work:-Re-evaluation of established thresholds for OHA and OHA/OHE via a retrospective analysis of more than 100.000 doping control samples analyzed in the Cologne doping control laboratory.-Re-evaluation of the established urinary concentration threshold for KT after implementation of a semi-quantitative approach for KT into the current ITP and inclusion of more than 5000 routine doping control samples.-Carbon isotope ratio determinations of KT, OHA, and KE as TCs and pregnanediol (PD) as an ERC in more than 100 samples exhibiting elevated urinary concentrations for KT in order to ensure that the elevated concentrations were not due to illicit steroid administrations and to enable the calculation of reference-based thresholds, especially for KT.-Investigation into an exploratory administration trial encompassing one male volunteer administered with 100 mg 11OXO to elucidate the best urinary marker for the detection in both the ITP and the confirmation procedure (CP) based on CIR.

## 2. Materials and Methods

### 2.1. Chemicals and Steroids

All solvents and reagents employed in this study were of analytical grade. Cyclohexane, tert-butyl methyl ether (TBME), methanol (MeOH), ethyl acetate, ethanethiol, ammonium iodide (NH4I), and acetonitrile (ACN) were from Merck (Darmstadt, Germany). Acetic anhydride employed in steroid derivatization was a blend of reagents purchased from Sigma Aldrich (Steinheim, Germany) and Merck and vacuum distilled in-house before use. Pyridine, sulphuric acid, and glacial acetic acid were also purchased from Sigma Aldrich. N-methyl-N-trimethylsilyltrifluoroacetamide (MSTFA) for silylation purposes was purchased from Chemische Fabrik Karl Bucher (Waldstetten, Germany). The β-glucuronidase from *Escherichia coli* employed for enzymatic hydrolysis was from Roche Diagnostics GmbH (Mannheim, Germany). Solid phase extraction (SPE) cartridges Chromabond^®^ C18 (500 mg, 6 mL) were obtained from Macherey & Nagel (Düren, Germany). Steroid reference material KT, KDHT, 17β-estradiol-diacetate (ESTRAc), and 3β-hydroxy-5α-androstane (RSTD) were purchased from Steraloids (Newport, RI, USA). 17α-Methyltestosterone (MeT), OHA, OHE, KA, KE, and PD were from Sigma Aldrich. The CO_2_ tank gas (Linde, Pullach, Germany) was calibrated against the secondary reference material USADA 33-1 provided by Cornell University (Ithaca, NY, USA) [12]. Helium (purity 5.0 and 4.6), argon (5.0), and oxygen (purity 5.0) were also from Linde.

### 2.2. Excretion Study Samples

One male volunteer administered 100 mg of 11OXO (2 capsules of a dietary supplement; purity and amount of substance checked via gas chromatography–mass spectrometry (GC/MS) and high-performance liquid chromatography–UV detection (HPLC-UV)). The CIR of 11OXO was determined to be at δ^13^C_VPDB_ = −29.3 ± 0.1‰ (*n* = 6). Three urine samples were collected before administration and twelve samples were collected during the two days following administration. All samples were stored frozen at −20 °C until analysis. The volunteer was healthy and sportive, practicing 2 to 4 h per week, and 48 years old, with a height of 180 cm and a weight of 84 kg. Considering the low number of volunteers, this excretion trial can only be considered as a proof-of-concept. The study was approved by the local ethics committee of the German Sport University Cologne (Number 084/2022) and written consent was given by the volunteer.

### 2.3. Sample Preparation for Quantification of Steroid Concentrations

In order to estimate the urinary concentrations of all relevant steroids, 2 mL of urine was processed by employing the Cologne routine doping control method [13,14]. In brief, after adjusting the pH to 7 by employing a phosphate buffer, β-glucuronidase was added to liberate the glucuronidated steroids. Free and formerly glucuronidated steroids were extracted via liquid–liquid extraction (LLE) using TBME, transferred to a new test tube, evaporated, and subjected to trimethylsilylation before injection into a gas chromatograph–triple quadrupole mass spectrometer (GC/MS/MS) [15,16]. System-related information is provided below.

### 2.4. Sample Preparation for Carbon Isotope Ratio Determinations

The already developed and validated approach for 11-hydroxy steroids was employed [11]. Between 1 and 10 mL of urine—depending on the urinary concentrations—was applied to a preconditioned solid-phase extraction cartridge, washed with water, and eluted with methanol. After evaporation, samples were dissolved in phosphate buffer (pH 7) and extracted with TBME to remove unconjugated steroids. Then, the samples were hydrolyzed at 50 °C for 60 min after adding β-glucuronidase. Finally, the formerly glucuronidated steroids were extracted via LLE, evaporated and transferred to autosampler vials, and subjected to high-performance liquid chromatography (HPLC)-based clean-up. Collected fractions were evaporated and acetylated by employing pyridine and acetic anhydride before a second HPLC clean-up step was employed, followed by CIR determinations.

### 2.5. Gas Chromatography—Combustion—Isotope Ratio Mass Spectrometry

All samples derived from the excretion study and selected samples derived from the athlete reference population to estimate urinary KT concentrations (vide infra) were injected into a Thermo Delta V Plus IRMS coupled to a Thermo Trace GC 1310 (Thermo Fisher, Bremen, Germany). The gas chromatograph (GC) was equipped with a TriPLus RSH Autosampler and connected to the IRMS by a GC IsoLink CNH combustion unit operated at 950 °C and a ConFlow interface (all ThermoFisher, Bremen, Germany). The employed GC column was an Agilent J&W Scientific DB-17MS (length 30 m, i.d. 0.25 mm, and film thickness 0.25 μm) from Agilent (Waldbronn, Germany). The temperature program started at 100 °C held for 2 min and then increased to 40 °C/min to 273 °C, then to 2 °C/min to 303 °C and finally again to 40 °C/min to 320 °C held for 5 min. Injections were performed in splitless mode by employing an Agilent Ultra Inert single taper liner held at 280 °C. The injection volume was 4 µL of sample together with 1 µL of RSTD in cyclohexane with a concentration of 40 µg/mL, and the constant helium flow was at 2 mL/min during the analytical run. Data were acquired and evaluated using Isodat 3.0 (ThermoFisher, Bremen, Germany). Low-resolution mass spectrometric data for peak purity assessment were acquired in parallel using a hyphenated Thermo ISQ single quadrupole mass spectrometer, which was connected to the GC column effluent employing a microchannel device (SGE, Sydney, Australia) and a deactivated restriction capillary (length 5 m, i.d 0.15 mm, SGE). The mass spectrometer was operated in electron ionization mode, and total ion current chromatograms were recorded from *m*/*z* 50 to 500 and evaluated using Thermo Xcalibur (version 2.2).

### 2.6. Gas Chromatography—Triple Quadrupole Mass Spectrometry

Quantification of KT was carried out on a Thermo TSQ8000 Evo coupled to a Trace GC 1310 (ThermoFisher, Bremen, Germany). Injections were performed in split mode with a split ratio of 1:15 at 300 °C by employing a Thermospray SSL Injector Module (ThermoFisher, Bremen, Germany). The injection volume was 1.8 µL and the GC column was an Agilent HP-Ultra with a length of 17 m, an inner diameter of 0.2 mm, and a film thickness of 0.11 µm. The temperature program started at 185 °C and was increased from 3 °C/min to 234 °C and then from 40 °C/min to 310 °C and held for 2 min before cooling. Helium was employed as carrier gas with a constant pressure of 17 psi. The temperature program and the constant pressure were adjusted as needed to maintain retention time stability. Data were acquired and evaluated using Thermo Xcalibur (version 2.2) and TraceFinder (version 5.0) software (ThermoFisher).

### 2.7. Gas Chromatography—High-Resolution/High-Accuracy Mass Spectrometry

All samples derived from the excretion study were analyzed on a Q Exactive GC Orbitrap GC/MS/MS system (Thermo Fisher) by employing high-resolution full scan data to search for potential new metabolites of 11OXO and to perform quantitation of all metabolites of interest. Samples derived from the excretion study were prepared as described for the quantification of steroids using 0.1 to 5 mL of urine. The analytical column and the temperature program were the same as for the triple quadrupole instrument described above. The transfer lines were set to 300 °C and 280 °C. The mass spectrometer acquired data in full scan mode with a scan range from *m*/*z* 70 to 700 at a resolution of 60,000. Data were collected and evaluated using Xcalibur (Version 4.0, ThermoFisher). Daily mass calibration of the instrument yielded mass accuracy in the range ± 2 ppm.

### 2.8. Athlete Reference Population to Reassess Urinary OHA and OHE Concentrations

To re-evaluate the already established thresholds for urinary concentrations of OHA and the diagnostic OHA/OHE ratio, *n* = 103,300 negative routine doping control samples analyzed in the Cologne laboratory in the years 2014 to 2018 were taken into consideration. Urinary concentrations for OHA and OHE were determined in parallel to the markers of the steroid profile by utilizing deuterated OHA as an internal standard together with an external calibration curve covering a linear range between 20 and 4000 ng/mL [13,14]. Concentrations for samples found above the linear range were estimated by using the highest calibrator as a single-point calibrator.

### 2.9. Athlete Reference Population to Estimate Urinary KT Concentrations

To re-evaluate the already suggested thresholds for the urinary concentrations of KT, *n* = 5232 routine doping control samples analyzed between April and June 2023 in the Cologne Doping Control Laboratory were considered. Within the population, 3674 (70%) male and 1558 (30%) female samples were included. The existing multiple reaction monitoring method was extended by two ion transitions suitable for KT, *m*/*z* 503.3 to *m*/*z* 413.3 as the quantifier and *m*/*z* 503.3 to *m*/*z* 323.3 as the qualifier, both at a collision energy of 7 eV. [13,14] Precursor ions were selected at a window width of 1.3 Da, and the retention time under the described chromatographic conditions was 15.35 min. A single-point calibrator urine sample at 20 ng/mL was injected every 20 samples to enable the semi-quantitative analysis of KT. In samples showing an elevated specific gravity (SG > 1.018), the concentrations for KT found were adjusted to an SG of 1.020 as recommended by the WADA [17].

### 2.10. Athlete Reference Subpopulation for IRMS-Based Investigations

Within the reference population investigated to establish a urinary concentration threshold for KT, several samples showed up with elevated concentrations of KT. The threshold was set arbitrarily to 70 ng/mL. All samples (*n* = 106, equivalent to 2% of all samples) were forwarded to IRMS in order to exclude elevated concentrations due to the illicit administration of KT or 11OXO and to enable the calculation of a population-based preliminary threshold for Δ-values built with KT.

### 2.11. Statistical Analysis

The Δ-values found were tested for Gaussian distribution via the Shapiro–Wilk test, and calculation of reference limits was conducted by adding the threefold standard deviation (SD) to the mean value in line with the recommendations of the International Federation of Clinical Chemists (IFCC) [18]. For urinary concentration thresholds, neither Gaussian nor log-normal distributions were detected, necessitating the implementation of a non-parametric approach to calculate potential reference-based limits. As in earlier publications, the far outside limit was chosen as the basis for threshold calculations [19,20,21,22]. A Wilcoxon rank-sum test with continuity correction was chosen to test for differences between male and female samples in the reference population for urinary KT concentrations. All calculations were carried out in Excel and R version 4.2.2. [23].

## 3. Results and Discussion

The results obtained within this research project will be presented in a logical order starting with the evaluation of the existing preliminary thresholds for the detection of 11OXO followed by the investigations into urinary KT and the attempt to establish potential thresholds here for both the urinary concentrations and CIR. These thresholds will be evaluated in light of the conducted excretion study after the administration of 11OXO.

### 3.1. Applicability of Preliminary Suggested Thresholds

The already published thresholds were based on a relatively small reference population of n = 85 athletes and it was suggested to establish population-based limits for the ITP considering urinary concentrations of OHA > 10.000 ng/mL and the ratio of OHA/OHA > 20 [9]. Samples above the defined limits should be forwarded to the IRMS for confirmation purposes.

Within this study, more than 100.000 doping control samples analyzed in the Cologne doping control laboratory between 2014 and 2018 were evaluated, and only three samples were found with elevated urinary concentrations of OHA, as shown in Table 1. In all three cases, the ratio of OHA/OHE found was also significantly elevated. Besides the high concentrations of OHA, other endogenous steroids were also found at high endogenous levels, especially androsterone (A). Sample S17 was not investigated by the IRMS as previously collected samples from the same athlete also showing elevated concentrations of A were already found to be unsuspicious, rendering the investigation of this sample unnecessary. Here, a medical condition was expected to be responsible for the generally elevated urinary concentrations of both A and OHA.

The second sample, S18_1, was forwarded to the IRMS due to the high A concentration and investigated via the routine doping control method, also focusing here on A itself, and it was found to be unsuspicious [24]. Another sample showing comparable concentrations for OHA and A (S18_2) was also forwarded to the IRMS, yielding a positive result, i.e., the CIR found for A and etiocholanolone (ETIO) demonstrated an exogenous origin for these steroids. Unfortunately, the CIR of OHA was measured in none of these samples, so it can only be speculated that, most probably, no 11OXO was administered here. The adverse finding demonstrated for S18_2 was most probably due to the administration of 4-androstenedione. Considering the extremely low prevalence of elevated OHA concentrations (3 out of 100.000), it is hardly possible to elucidate if the previously established threshold is useful in doping controls or not.

Investigating the relevant concentration ratio OHA/OHE provided a different picture (Table 2). Approximately 10% of all routine doping control samples showed OHA/OHE ratios above 20. This high number of potentially suspicious samples prevented the forwarding of samples to the IRMS routinely, considering the high workload involved in IRMS sample preparation. Even increasing the threshold to a certain extent would not help to solve the issue as the distribution of the ratios found was quite broad. This is demonstrated exemplarily by the data analyzed in 2018 and shown in Figure 1. Even doubling the threshold to OHA/OHE > 40 would still result in more than 300 suspicious samples per year. Considering the presumed low prevalence of 11OXO misuse makes the effort necessary to analyze such a high number of samples with IRMS at least questionable.

### 3.2. Urinary Concentrations Found for KT

Found concentrations for urinary KT found showed a broad distribution and were neither Gaussian nor log-normal-distributed, as demonstrated in Figure 2. In particular, the relatively high number of samples showing very low concentrations impeded a straightforward parametric approach for the calculation of a reference limit. Approximately one-fifth of all samples showed urinary concentrations below 2 ng/mL. Therefore, the non-parametric “twice the far outside limit” was calculated by adding six times the inter-quartile range to the 75% quartile [20,21,22]. This procedure resulted in a threshold for urinary KT of 130 ng/mL. Doping control samples above this threshold should be further investigated by the IRMS to differentiate between naturally elevated concentrations and doping offenses. Within the investigated population of more than 5000 samples, only 9 (<0.2%) showed concentrations above the established threshold with a maximum concentration of 204 ng/mL. Therefore, the suggested threshold is presumed to result in a reasonable number of samples forwarded to the IRMS per year. For comparison, between 2 and 3% of all routine doping control samples are forwarded to the IRMS based on the longitudinal steroid profiling in the context of testosterone doping [25,26].

Although a significant difference was noted between the median of the male (10.9 ng/mL) and female (8.6 ng/mL) samples, resulting in a potentially slightly lower threshold for the female sample of 110 ng/mL, it was decided to suggest one threshold applicable to all samples for the sake of simplicity.

### 3.3. Carbon Isotope Ratios Found for KT and Its Metabolites

All samples showing elevated urinary concentrations of KT (>70 ng/mL) were forwarded to the IRMS in order to exclude suspicious samples from the population and to enable the calculation of appropriate population-based reference limits for KT and its main metabolites, OHA and KE. Investigating not only KT but also OHA and KE allowed for the identification of potentially suspicious samples as, here, population-based CIRs were already published [24,27].

The investigated δ-values showed a broad distribution from −16 to −24‰, as shown in Figure 3. Therefore, the investigated population encompassed all possible CIRs expected in an athlete population, and the calculated Δ-values can be straightforwardly applied to doping control samples [28,29].

The distribution of all Δ-values was found to be Gaussian (Figure 4), which enabled the calculation of corresponding thresholds straightforwardly through the addition of the three-fold standard deviation to the mean value, resulting in the 99.7% reference limit. The calculated values are summarized in Table 3. A slightly higher percentage of male samples (82% compared to 70% in the underlying population investigated regarding urinary concentrations) was forwarded to the IRMS, but within the Δ-values found, no significant difference between the male and female samples was identified. This finding corroborates earlier results demonstrating the uniform distribution of Δ-values among sexes [21,30].

In general, slightly depleted values were found for KT compared to its metabolites OHA and KE. This fact together with the larger SD found for KT resulted in the higher population-based reference limit for PD-KT. Increased measurement uncertainty for KT was also noted within the repeated preparations of the blank urine sample (*n* = 11), with each batch of samples investigated. The determined mean values and SDs are listed in Table 4.

### 3.4. 11OXO Administration Trial

In order to test the herein-established thresholds for urinary concentrations of KT and CIR and to enable a comparison to the already published limits, an administration study after the oral application of 100 mg 11OXO was examined.

#### 3.4.1. Urinary Concentrations

Directly after the administration of 11OXO, all investigated metabolites showed a significant increase in urinary concentrations, as depicted in Figure 5. Maximum concentrations were reached 3 or 5.5 h after application, followed by a steep decrease over the following hours. The values remained slightly elevated until 33 h after administration but already returned to normal endogenous ranges after 10 to 14 h. Considering the preliminarily established threshold for OHA, only three samples were found suspicious 3 to 7.5 h after application. The ratio of OHA/OHE never reached the suggested threshold of 20 but remained below 8 all the time [9]. Regarding the other known metabolites of 11OXO, KE in particular showed a strong increase in urinary concentrations from around 100 ng/mL to more than 330 µg/mL.

Urinary KT was demonstrated to be a direct metabolite of 11OXO and was strongly influenced after administration with an increase in urinary concentrations from below 10 ng/mL to more than 70 µg/mL. Concentrations were found above the threshold from 1 to 14 h after administration.

In a recently conducted excretion study administering KT, neither urinary concentrations of OHA nor the ratio of OHA/OHE showed a significant increase [11].

#### 3.4.2. Urinary CIR and Δ-Values

The CIRs of all investigated metabolites of 11OXO were found to be strongly depleted directly after administration. In general, the metabolites reflected the CIR of the administered 11OXO and showed values around −29‰. Only KT was less depleted with minimum values around −27.5‰. This behavior is also reflected in the corresponding Δ-values as shown in Figure 6. The employed ERC (PD) was determined to have a CIR around −22.8‰, resulting in maximal Δ-values between 5 and 6‰ for OHA and especially KE, while PD-KT barely reached the established threshold of 4.6‰.

Considering the population-based thresholds found in this work, PD-KE seems to be the most promising parameter to confirm the misuse of 11OXO, proving the exogenous origin of KE for 33 h after administration. But also, OHA and OHE resulted in comparable detection windows of 21 and 25 h, respectively. KT itself was found to be of lower value during the IRMS-based confirmation, mainly as the influence of the administered 11OXO seemed to be diminished. Considering the strongly elevated urinary concentrations may suggest an ongoing isotopic fractionation in the direction of KT, which was corroborated by the measured CIR values for KDHT also showing minimal values around −27‰ [11]. Here, further investigations may be necessary considering the underlying human steroid metabolism, especially as after the administration of KT, urinary KT directly reflects the CIR of the administered compound [11].

### 3.5. Potentially New Metabolites

A careful inspection of the high-resolution/high-accuracy mass spectra after administration did not enable the detection of new metabolites of 11OXO. Interestingly, one metabolite described after the administration of KT was also found to be significantly elevated here [11]. The metabolite described as M606 increased (in its ratio built with the internal standard) by a factor of roughly 100, but it decreased as fast as the other investigated metabolites.

## 4. Conclusions

Adrenosterone is prohibited at all times for competing athletes and it is listed on WADA’s Prohibited List as an anabolic androgenic steroid. Within this research project, the current approach to detect its misuse was reinvestigated using a retrospective data analysis of more than 100.000 routine doping control samples analyzed in the Cologne doping control laboratory in the years 2014 to 2018. While samples showing elevated urinary concentrations of OHA beyond the established threshold of 10,000 ng/mL were rare, the number of samples showing an elevated OHA/OHE ratio was surprisingly high. Based on these findings, it was hypothesized that both markers may not be suitable to detect 11OXO administrations in a reliable way, and a recently investigated new steroid—KT—was considered as a novel tool to detect 11OXO misuse in both the ITP and CP. The ITP is based on urinary concentrations and considering more than 5000 routine doping control samples enabled us to establish a threshold for KT at 130 ng/mL. Urinary concentrations of KT found below this threshold can be considered to be of endogenous origin, concentrations above this threshold may occur and should be further investigated by the IRMS-based CP. More than 100 samples with urinary KT concentrations above 70 ng/mL were investigated, and all measured CIRs demonstrated to be of endogenous origin. Based on this population, it was possible to calculate population-based reference limits for Δ-values using PD as an ERC. All Δ-values fell in the expected range and interestingly, KT itself was demonstrated to be less suitable as a marker for the administration of 11OXO compared to other metabolites, especially KE. As many WADA-accredited doping control laboratories worldwide already have IRMS methods in place using KE or OHA as ERCs, the implementation of criteria to detect 11OXO administrations will be straightforward. Of course, it will have to be considered that KE and OHA are currently implemented as ERCs to detect the misuse of testosterone and may not be applicable for this purpose if the sample is suspicious for the administration of 11OXO or KT itself. Interpretations of IRMS results will have to consider the ITP findings in order to enable a reasonable evaluation of each doping control specimen. Nevertheless, implementing KT in the ITP will enable the identification of samples suspicious for the administration of 11OXO and KT itself, and the IRMS-based confirmation will allow for differentiation between the endogenous or exogenous origin of urinary metabolites.

## Figures and Tables

**Figure 1 metabolites-14-00141-f001:**
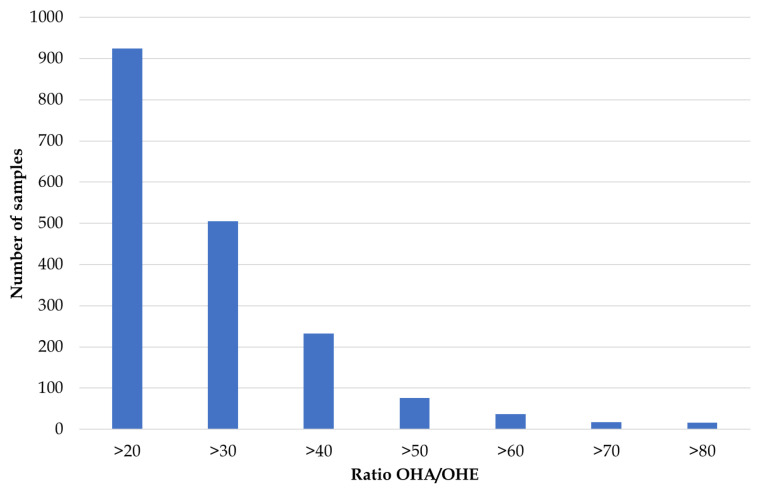
Number of samples showing elevated ratios of OHA/OHE in the subset of *n* = 21,000 routine doping control samples analyzed in Cologne in 2018.

**Figure 2 metabolites-14-00141-f002:**
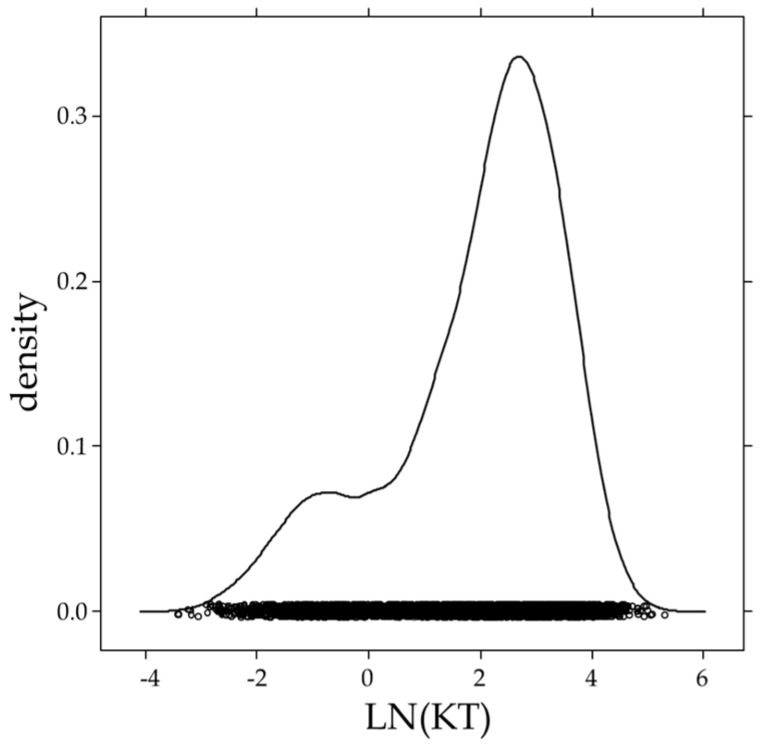
Urinary concentrations of 11-keto-testosterone (KT) found in the investigated athlete reference population (*n* = 5232) by employing logarithmic transformation.

**Figure 3 metabolites-14-00141-f003:**
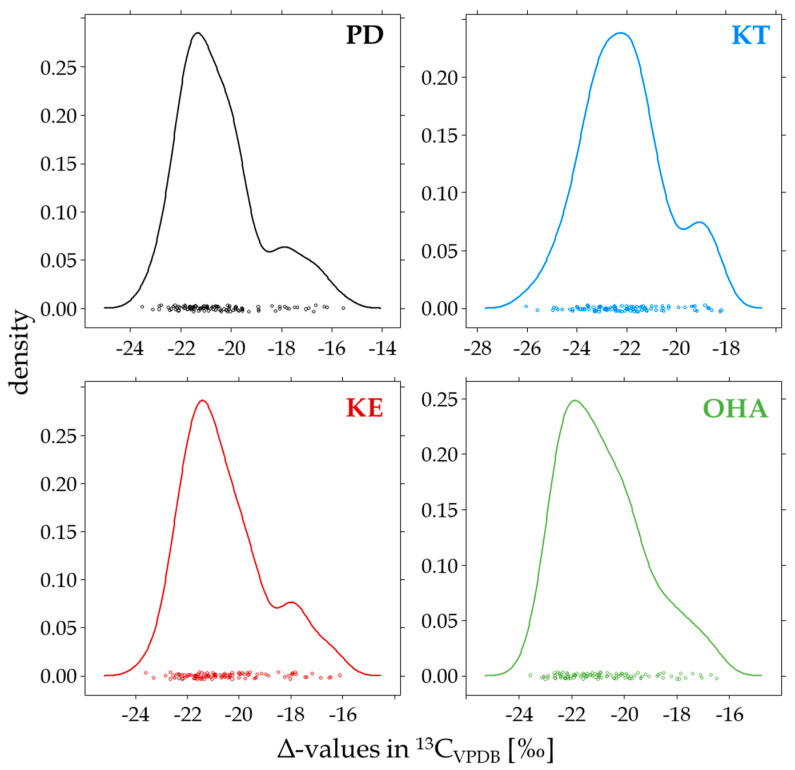
Carbon isotope ratios measured in the subpopulation of *n* = 106 samples for the endogenous reference compound pregnanediol (PD) and the target analytes 11-keto-testosterone (KT), 11-oxo-etiocholanolone (KE), and 11-hydroxy-androsterone (OHA).

**Figure 4 metabolites-14-00141-f004:**
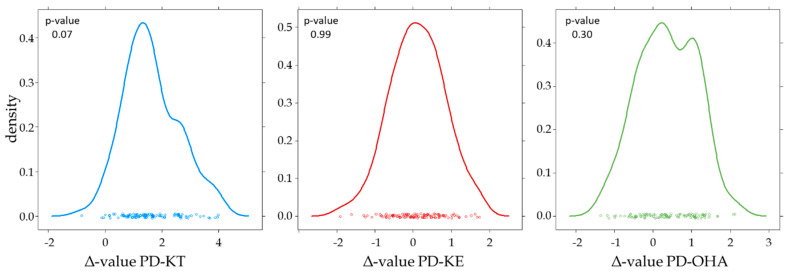
Distribution of Δ-values found in the investigated subpopulation of *n* = 106 samples.

**Figure 5 metabolites-14-00141-f005:**
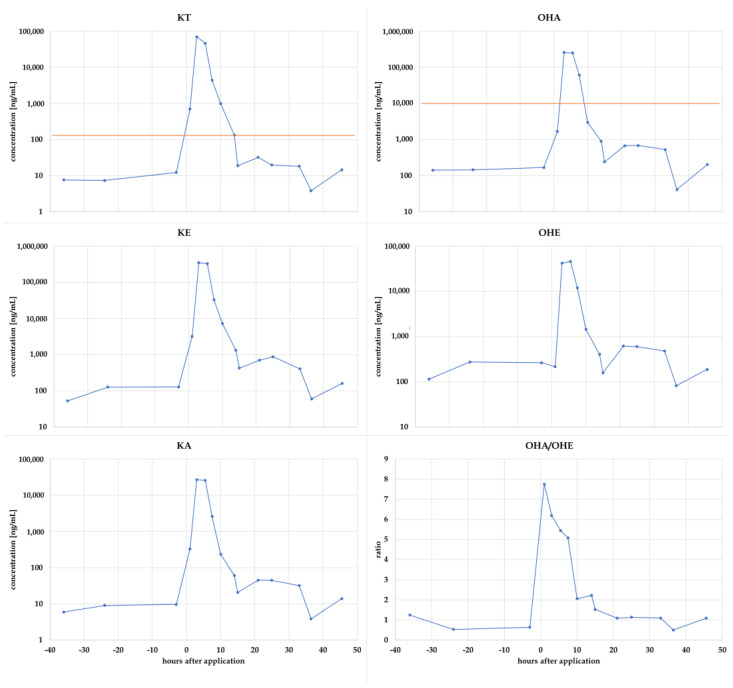
Urinary concentrations of KT, OHA, OHE, KE, and KA measured after the oral administration of 100 mg 11OXO. Orange lines represent the current concentrations thresholds for KT (this work) and OHA [9]. Additionally, the ratio of OHA/OHE is shown (**bottom right**).

**Figure 6 metabolites-14-00141-f006:**
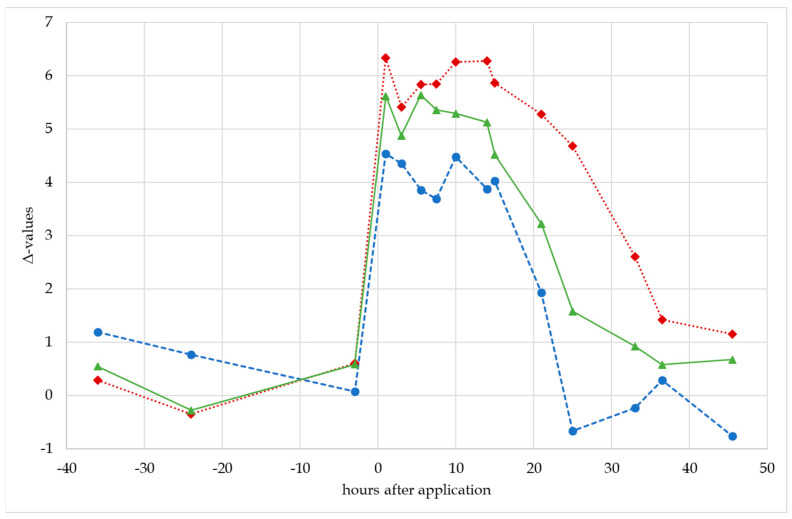
Δ-values found after the oral administration of 100 mg 11OXO. Blue circles represent PD-KT, green triangles represent PD-OHA, and red diamonds represent PD-KE.

**Table 1 metabolites-14-00141-t001:** Steroid profile data found within 3 samples showing elevated urinary concentrations of OHA. Further information in the text.

Sample	OHA [ng/mL]	OHE [ng/mL]	OHA/ OHE	A [ng/mL]	ETIO [ng/mL]	PD [ng/mL]	SG	Sex	IRMS
S17	10,465	147	71.2	22,000	3800	1700	1.025	female	NA
S18_1	25,250	606	41.7	22,000	6100	1700	1.029	female	neg
S18_2	25,000	691	36.2	27,000	15,000	250	1.032	female	pos

**Table 2 metabolites-14-00141-t002:** OHA/OHE ratios found in doping control samples in the years 2014 to 2018.

Year	*n*	OHA/OHE > 20	%
2014	17,800	1837	10.3
2015	19,500	1945	10.0
2016	23,000	2246	9.8
2017	22,000	2101	9.6
2018	21,000	1808	8.6

**Table 3 metabolites-14-00141-t003:** Δ-values and calculated population-based thresholds found in the investigated subpopulation of *n* = 106 athletes.

	PD-KT	PD-KE	PD-OHA
mean	1.6	0.1	0.4
SD	0.99	0.73	0.76
Limit	4.6	2.3	2.7

**Table 4 metabolites-14-00141-t004:** Mean δ-values and calculated standard deviations found for the repeated preparation (*n* = 11) of a blank urine sample.

	KT	KE	OHA	PD
mean	−23.9	−21.7	−22.3	−21.9
SD	0.61	0.48	0.59	0.41

## Data Availability

The raw data supporting the conclusions of this article will be made available by the authors upon reasonable request.

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
