# Peer review of "Employing 11-Ketotestosterone as a Target Analyte for Adrenosterone (11OXO) Administration in Doping Controls"

_metabolites, 2024, doi:10.3390/metabo14030141_

Round 1

Reviewer 1 Report

Comments and Suggestions for Authors

The manuscript is devoted to the improvement of doping control system and proposes 11-oxo-testosterone (KT) as a reliable marker of adrenosterone (11OXO) administration. The authors performed the comprehensive and impressive study involving a number of analytical techniques. The obtained results are clearly presented and properly justified. The manuscript makes very good impression and is of undoubted interest to readers. I would like to congratulate the authors on their good job and recommend the manuscript for publication after correcting some minor issues:

Please, provide the structural formulas of the studied steroids in the manuscript.

The abstract section would benefit from the brief description of the used analytical techniques.

There are some typos in the manuscript.

Author Response

Dear reviewer,

thank you very much for your kind and constructive comments.

Of course, we will add the steroidal structures. As these are very common and well known in the field of sports drug testing we would prefer to add them as Supplementary Material.

The analytical techniques have been added to the abstract and we tried to carefully search for all typos. 

Reviewer 2 Report

Comments and Suggestions for Authors

The manuscript titled "Employing 11-Ketotestosterone as a Marker for Adrenosterone (11OXO) Administrations in Doping Controls" presents a thorough investigation into the detection of 11-oxo-androsterone (11OXO) abuse in doping control settings. By reassessing established thresholds for urinary metabolites and suggesting the use of 11-ketotestosterone (KT) as an indicator of 11OXO administration, the study offers significant contributions to doping control methodologies.

The strength of the study lies in its extensive analysis of over 100,000 routine doping control samples, providing a robust foundation for evaluating proposed thresholds. Particularly noteworthy is the proposal of a urinary concentration threshold of 130 ng/mL for KT, supported by a substantial dataset, which aids in effectively identifying suspicious samples. Additionally, the inclusion of carbon isotope ratio (CIR) determinations enhances the reliability of detection methods, offering valuable insights into distinguishing between endogenous and exogenous steroid use. However, several limitations warrant consideration:

 1.      While the study analyzes a large number of routine doping control samples, the representativeness of the sample population needs careful assessment. The generalizability of findings may be limited if the sample population does not adequately reflect the diversity of athletes and doping practices worldwide.

 2.      The establishment of thresholds, such as the proposed urinary concentration threshold for KT, is pivotal for effective detection. However, the sensitivity and specificity of these thresholds may vary due to factors like assay precision, metabolite stability, and individual variability. Further validation studies or sensitivity analyses could bolster confidence in the proposed thresholds.

 3.      The manuscript acknowledges the challenge of distinguishing between naturally elevated concentrations and doping offenses. While the proposed thresholds aim to minimize false positives, there remains a risk of false negatives, where genuine instances of doping may go undetected. Exploring strategies to mitigate both types of errors would enhance the reliability of the detection methodology.

Comments on the Quality of English Language

The quality of English language in the manuscript is generally good.

Author Response

Dear reviewer,

Thank you very much for your helpful comments. Please find our answers below:

  1. While the study analyzes a large number of routine doping control samples, the representativeness of the sample population needs careful assessment. The generalizability of findings may be limited if the sample population does not adequately reflect the diversity of athletes and doping practices worldwide.

Routine doping control samples analysed in the Cologne Doping Control Laboratory are derived from all over the world – so in general the investigated population should represent a sub-population of all athletes. Of course, a bias towards European and especially German Athletes can be expected. But considering the compounds under investigation here, no scientific data is available supporting a significant difference in metabolism depending on the diversity of athletes. Future research and/or the application of the suggested threshold world-wide may shed more light on potential differences.  

  1. The establishment of thresholds, such as the proposed urinary concentration threshold for KT, is pivotal for effective detection. However, the sensitivity and specificity of these thresholds may vary due to factors like assay precision, metabolite stability, and individual variability. Further validation studies or sensitivity analyses could bolster confidence in the proposed thresholds.

We fully agree that the investigations carried out so far can only be estimated as the starting point and that additional data derived over the years will be important to corroborate (our revise) our findings. An important point here is the application of two different methodologies for initial screening and confirmation purpose. So even if the proposed threshold may not be appropriate, we can exclude that any athlete´s sample may produce a false positive result. This is of outmost importance.

  1. The manuscript acknowledges the challenge of distinguishing between naturally elevated concentrations and doping offenses. While the proposed thresholds aim to minimize false positives, there remains a risk of false negatives, where genuine instances of doping may go undetected. Exploring strategies to mitigate both types of errors would enhance the reliability of the detection methodology.

Yes, but as mentioned before, our main concern is to avoid false positive cases. False negative cases are also problematic, but can be accepted at this stage of research. If more data may be available, thresholds can be evaluated accordingly.

Reviewer 3 Report

Comments and Suggestions for Authors

This manuscript discusses the prohibited substance Adrenosterone (11OXO) in sports, emphasizing its detection through monitoring urinary concentrations of its main metabolites. The existing thresholds for sports drug testing are problematic due to naturally elevated concentrations. The study explored the metabolism of 11-oxo-testosterone (KT) and proposed a new urinary concentration threshold for KT, determined through a reference population analysis of routine doping control samples. The suggested threshold for KT is proposed to be 130 ng/mL. The research aims to enhance the analytical approach for detecting the administration of Adrenosterone in anti-doping efforts. I find this finding interesting and could potentially impact doping controls, therefore I suggest accepting this paper. 

Author Response

Dear reviewer,

thank you very much for your comment.

Reviewer 4 Report

Comments and Suggestions for Authors

Firstly, I would like to congratulate the authors of the manuscript on their very interesting study on

“Employing 11-Ketotestosterone as Target Analyte for Adrenosterone 11OXO) Administrations in Doping Controls”. Primarily, the sample number for excretion trial seems low, how will authors justifies the trial results which are from a single individual. Please add this justification in your manuscript.  In addition to this there are few minor observations that the authors should address before final submission.

1.            Avoid abbreviations at the start of a sentence.

2.       The excretion trial on a single volunteer seems a very low sample number.  

3.            Define abbreviations at first citation, throughout the manuscript.

4.      Add full stop at the end of each sentence after reference in text.

Best of Luck. 

Author Response

Dear reviewer,

Thank you very much for your helpful comments.

Employing an excretion trial with only one volunteer can of course only be considered as a proof-of-concept demonstrating that the established thresholds in principle will work. We emphasized this point in our article.

  1. Avoid abbreviations at the start of a sentence.

            Avoided

  1. The excretion trial on a single volunteer seems a very low sample number.

            Yes, we clarified that this can only be considered as a proof-of-concept.  

  1. Define abbreviations at first citation, throughout the manuscript.

            The article was carefully revised.

  1. Add full stop at the end of each sentence after reference in text.

            Done